# Acupuncture for military veterans with posttraumatic stress disorder and related symptoms after combat exposure: Protocol for a scoping review of clinical studies

Hye-Bin Seung[1☯], Jungtae Leem[2☯], Hui-Yong Kwak[3], Chan-Young Kwon[4], Sang-Ho Kim[5] *

1 College of Korean Medicine, Daegu Haany University, Gyeongsan-si, Gyeongsangbuk-do, Republic of Korea, 2 College of Korean Medicine, Wonkwang University, Iksan, Republic of Korea, 3 Republic of Korea Army, Capital Defense Command, Gwacheon-daero, Gwanak-gu, Seoul, Republic of Korea, 4 Department of Oriental Neuropsychiatry, Dong-Eui University College of Korean Medicine, Busan, Korea, 5 Department of Neuropsychiatry of Korean Medicine, Pohang Korean Medicine Hospital Affiliated to Daegu Haany University, Pohang-si, Gyeongsangbuk-do, Republic of Korea

☯ These authors contributed equally to this work.
* omed22@naver.com

**Data Availability Statement:** No datasets were generated or analysed during the current study. All

## Abstract

Posttraumatic stress disorder is caused by traumatic events such as death, serious injury, and sexual violence. Military personnel and veterans are at high risk for posttraumatic stress disorder. Conventional posttraumatic stress disorder treatments have certain limitations. Complementary and integrative medicine treatments, especially acupuncture, are potential novel first-line treatments that may overcome these limitations. We aim to investigate the current status of the available clinical evidence related to acupuncture treatment for posttraumatic stress disorder in war veterans. We will follow the scoping review process as previously described. The study question is as follows: "Which types of clinical research designs, study types, study durations, adverse events, and clinical outcomes have been reported regarding acupuncture therapy for posttraumatic stress disorder in military veterans?" We will perform a comprehensive search of Medline, Excerpta Medica dataBASE, Cochrane Central Register of Controlled Trials, Web of Science, Scopus databases, Allied and Complementary Medicine Database, Cumulative Index to Nursing and Allied Health Literature, and PsycArticles databases, as well as Chinese, Korean, and Japanese databases, from inception to June 2022. Data from the included studies will be collected and descriptively analyzed in relation to our research question. The extracted data will be collated, synthesized, and summarized according to the analytical framework of a scoping review. The protocol of this study adheres to the Preferred Reporting Items for Systematic Reviews and Meta-Analyses Extension for Scoping Reviews to ensure the clarity and completeness of our reporting in all phases of this scoping review (Protocol registration: https://osf.io/t723f/). The findings of this scoping review will provide fundamental data that will help researchers identify appropriate research questions and design further studies on the use of

relevant data from this study will be made available upon study completion.

**Funding:** This work was supported by the National Research Foundation of Korea (NRF) grant funded by the Korean government (MSIT) (No. 2021R1F1A105928211). SHK has received this fund. The funding source had no input in the interpretation or publication of the study results. The funders had no role in study design, data collection and analysis, decision to publish, or preparation of the manuscript. ※ MSIT: Ministry of Science and ICT.

**Competing interests:** The authors have declared that no competing interests exist.

acupuncture for PTSD management in military veterans. These results will be helpful for developing disaster site-specific research protocols for future clinical trials on this topic.

## Introduction

Posttraumatic stress disorder (PTSD) occurs in individuals who have been exposed to one or more traumatic events such as death, serious injury, or sexual violence [1]. The clinical presentation of PTSD includes fear-based reexperiencing, emotional and behavioral symptoms, anhedonic or dysphoric mood states, and dissociative symptoms [1]. Active-duty military personnel and veterans often experience or witness terrorist attacks, violent crimes and abuse, natural disasters, serious accidents, or violent personal assaults [2], making them extremely vulnerable to PTSD [3]. Lifetime prevalence of PTSD has been estimated from 1.3%–8.8% [4]. Among Vietnam War veterans, 30.9% were diagnosed with PTSD [5]. In addition, 12.9% of Iraq veterans and 7.1% of Afghanistan veterans reportedly have PTSD [6]. PTSD is usually associated with comorbidities such as depression, anxiety, and substance use disorders [7]. In veterans with PTSD, suicidal ideation was reportedly 23.8% and suicide attempts were 6.8% owing to military combat trauma [8]. Furthermore, PTSD is related to chronic pain [9], cardiovascular disease [10], metabolic syndrome, and elevated C-reactive protein levels [11]. Many military personnel from around the world have been on the battlefield in civil and local wars and counterterrorism [12]. The conflict in Ukraine started in Donbas in 2014 when Russia annexed Crimea and has recently escalated to an all-out war. A study has already been published examining PTSD caused by the war in Donbas [13]. Furthermore, enormous mental health consequences are expected for the people of Ukraine [14].

Most clinical guidelines recommend cognitive behavioral therapy (CBT) and selective serotonin reuptake inhibitors as first-line psychological and pharmacological therapies, respectively, for patients with PTSD [15]. However, CBT has a limitation in that it is time consuming and costly, requires training to perform treatments [16], and is difficult to use owing to low accessibility [17]. Although internet-based CBT may be an option to increase access to treatment, it may not be suitable for patients who are uncomfortable with technology [18]. Only two medications have been approved by the Food and Drug Administration for the treatment of PTSD in adults (sertraline and paroxetine) [19]. Selective serotonin reuptake inhibitors inhibit cytochrome P450 enzymes in the liver and may cause several adverse effects such as extrapyramidal symptoms, serotonin syndrome, QT prolongation, congenital malformations, and hyponatremia [20]. Stigma exists in military culture, along with treatment barriers such as costs and low accessibility to mental health care [21]. PTSD tends to be more prominent among personnel who meet the criteria for a mental health problem [22].

Recently, complementary and integrative medicine treatments for patients with PTSD have gained increased attention [23]. Acupuncture is a nonpharmacological therapy involving the insertion of needles into specific points on the body or ear, known as acupuncture points. The use of acupuncture in veterans is a promising complementary and integrative medicine approach [24]. Specific forms of auricular acupuncture, such as the National Acupuncture Detoxification Association (NADA) protocol and battlefield acupuncture (BFA), are used in the health management of military personnel. The NADA protocol, which consists of five standard auricular acupuncture points, was originally designed to help acute heroin addiction, although it is also used for a broad range of mental health problems [25]. The NADA protocol was effective in improving PTSD-induced insomnia in combat veterans [26]. BFA, which is a safe, fast, and easily applied acupuncture treatment, has been proven effective for treating a

variety of pain conditions [27]. Evidence suggests that BFA can help with chronic pain and sleep disturbances in veterans [28]. Sleep disturbances are a core feature of PTSD and can affect its progression [29]. Furthermore, chronic pain in veterans with PTSD is associated with many negative health-related outcomes, such as disability, depression, sleep disturbance, catastrophizing beliefs, and lower function [30]. Therefore, although BFA has mainly been used for pain, it has the potential to alleviate PTSD symptoms in veterans through pain control and improved sleep.

A scoping review is more appropriate than a systematic review when the research explores a broad range of questions, such as which research design was adopted in the topic of interest, and can help to identify major concepts and characteristics in the existing literature as well as knowledge gaps [31]. As studies on acupuncture for PTSD in veterans have not been actively conducted, our research team determined that a scoping review method with a wider view of the relevant field may be more appropriate for this study than a systematic review of randomized controlled studies.

We aim to explore the clinical research designs used in studies conducted on acupuncture treatment for war veterans with PTSD. We will also focus on detailed methodological characteristics, such as treatment regimens, participant characteristics, and frequently used outcomes. This study will enable us to identify any research gaps between clinical studies and the needs of physicians regarding clinical evidence. The results of our scoping review can be used as fundamental data for identifying the appropriate research questions in research planning of future clinical studies and systematic reviews.

## Materials and methods

### Study design and registration

We follow the scoping review process described by Arksey and O'Malley [32], Levac et al., and Tricco et al. [33,34] as well as the Preferred Reporting Items for Systematic Reviews and Meta-Analyses Extension for Scoping Reviews criteria. This review protocol was registered with the Open Science Framework (https://osf.io/t723f/) on July 23, 2022.

### Stage 1: Identifying the study questions

This stage is based on a preliminary literature search of published clinical evidence. Agreement among the research team members was required to obtain better scoping review research questions. The research team comprises three specialists in neuropsychiatry (SHK, CYK, and HYK), a specialist in clinical research on traditional East Asian medicine (JL), and an undergraduate researcher (HBS). All members of the research team agreed on the concepts and revisions of the research questions. In our scoping review, we will attempt to answer the following questions:

1. What are the characteristics (e.g., research design and target population) of studies on the use of acupuncture for PTSD management in military veterans?

2. Which clinical outcomes were adopted in previous studies on PTSD management in military veterans?

3. What is the regimen of acupuncture therapy for PTSD management in military veterans?

4. What have previous studies reported on with respect to the effectiveness and safety of using acupuncture for treating PTSD in military veterans?

## Stage 2: Identifying relevant studies

**Information sources.**    We will restrict this review to peer-reviewed studies on acupuncture for PTSD. A literature search will be conducted from inception to June 2022. The following databases will be searched: Medline (via PubMed), Excerpta Medica dataBASE, Cochrane Central Register of Controlled Trials, Web of Science, Scopus, Allied and Complementary Medicine Database, Cumulative Index to Nursing and Allied Health Literature, PsycArticles, China National Knowledge Infrastructure, Wanfang, VIP, Oriental Medicine Advanced Searching Integrated System, Korea Citation Index, and Citation Information by NII. We will also consider gray literature searches such as conference proceedings and doctoral dissertations using Google Scholar. The reference lists of the relevant reviews and retrieved articles will be manually searched. Attempts will be made to contact the authors of published papers for which electronic files cannot be accessed. This search strategy was developed through consultation with a clinical researcher, an expert on literature review, and a specialist on psychiatric diseases. Search terms will comprise disease (PTSD due to war exposure) and intervention terms (acupuncture). A combination of various synonyms and related medical subject headings will be used in the search strategy. The search terms and strategies are detailed in S1 Table.

**Eligibility criteria: Study types.**    Clinical research studies examining the effects of acupuncture in patients with PTSD (military veterans) will be included. The following types of clinical research studies will be included in this review: randomized controlled clinical trials, quasi-randomized controlled trials, nonrandomized controlled trials, single-arm trials, case series, cross-sectional studies, and feasibility studies

We will exclude case reports with less than three patients [35], literature reviews, and preclinical studies. Moreover, we will review the reference articles of each systematic review.

**Eligibility criteria: Types of participants.**    Military veterans with PTSD and related symptoms after combat exposure will be included. We will include studies that used standardized diagnostic criteria for PTSD (such as those presented in the Diagnostic and Statistical Manual of Mental Disorders and the International Classification of Diseases). We will also include studies that used cut-off values from validated PTSD evaluation tools (such as the PTSD Checklist, Impact of Event Scale-Revised, and Clinician-Administered PTSD Scale) as the inclusion criteria for participants. Furthermore, we will include PTSD with comorbid medical illnesses such as tinnitus or traumatic brain injury.

**Eligibility criteria: Intervention types.**    Various acupuncture therapies include manual acupuncture, electroacupuncture, bee-venom acupuncture, pharmacopuncture, warm-needle acupuncture, fire needle acupuncture, and acupotomy. However, we will not include acupressure therapy in this study, and we will not restrict the concomitant treatment. With the exception of East Asian traditional medicine interventions, such as herbal medicine, moxibustion, cupping, and tui-na, any type of control group intervention will be included. The treatment period, dosage, and acupuncture treatment frequency will not be restricted.

**Eligibility criteria: Outcome measurements.**    We will also consider various symptoms after the diagnosis of PTSD. We will not restrict the types of outcome variables. The outcomes will be categorized according to a previous study [36] as follows: 1) outcomes related to psychological aspects (anxiety, fear, anger, irritability, guilt, shame, apathy, distrust, sadness, frustration, alienation, loss of confidence, and mourning); 2) outcomes related to somatic aspects (insomnia, palpitation, pain, anorexia, and fatigue); and 3) outcomes related to cognitive aspects (decreased memory, difficulty making decisions, repeated recall of traumatic events, and difficulty concentrating). In terms of safety, we will also investigate the incidence of adverse events and dropout rates.

### Stage 3: Study selection

The inclusion and exclusion criteria were developed with the consensus of the research team. Two reviewers (HBS and HYK) will independently conduct the article selection process. After eliminating duplicate publications in the first phase, the titles and abstracts of the screened articles will be reviewed for inclusion. For potential articles in the second screening phase, the full text will be reviewed to determine whether the article meets the inclusion criteria. The reasons for inclusion and exclusion will be recorded for each article according to the predetermined criteria. In cases of discrepancies, an agreement will be achieved through mediation by the independent researcher. Details of the study selection process are shown in Fig 1.

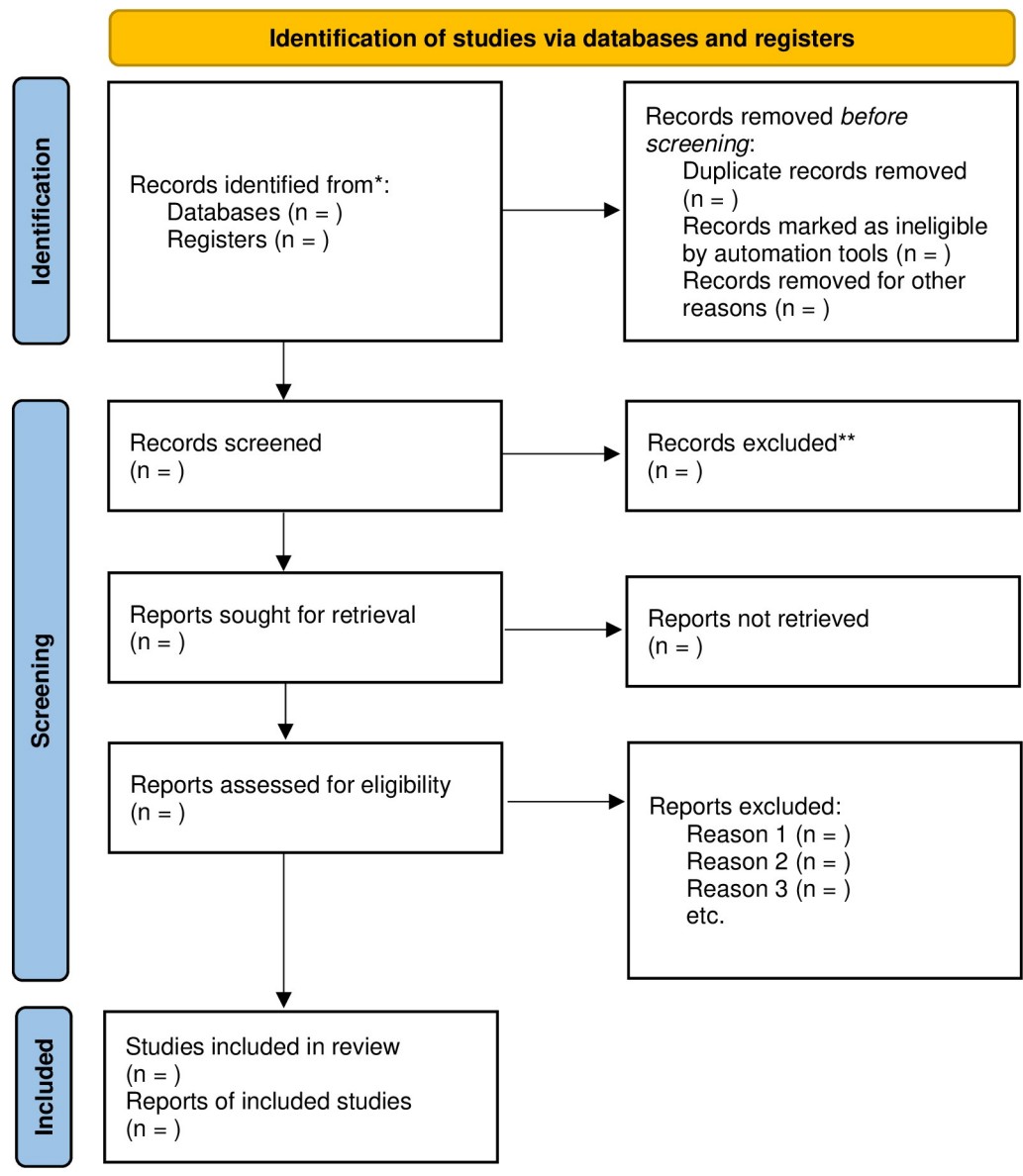

**Fig 1. Flowchart of identification and screening for the eligible studies.**

### Stage 4: Charting the data

A data extraction sheet was prepared by the study team after the pilot test. After multiple iterations, a standard data extraction form was created. The following items will be collected from the included studies: 1) general information, such as the first author's name, country, publication year, and research design; 2) participants' demographic data, such as age, sex, number of participants (initial and final), diagnostic criteria, and disease duration and severity; 3) detailed information on interventions such as type of acupuncture, acupuncture points, treatment dosage (number, frequency), treatment period, and details of control/concomitant interventions; and 4) outcome variables. Data regarding effects, safety, and research findings will also be extracted.

Two reviewers (HBS and HYK) will separately undertake the data extraction using the described method and will crosscheck the data from all included articles. Any disagreement will be resolved through discussion until consensus with a third researcher (SHK).

### Stage 5: Collating, summarizing, and reporting the results

The extracted data will be collated, synthesized, and summarized according to the analytical framework of the scoping review.

In the qualitative analysis stage, we will provide information on the included studies, including the author's name, country, publication year, combat details, number of participants, sex, age, research design, and type of treatment/control group interventions. In a second table titled "Detailed information of acupuncture treatment," we will provide detailed information regarding interventions such as type of acupuncture, location of acupuncture points, number of treatments, treatment frequency, treatment period, and details of control/concomitant interventions. In a third table titled "Effects and safety of acupuncture treatment for PTSD and related symptoms," the results of every outcome from the included studies will be presented along with the conclusion of each study. The number of adverse events will also be presented. In addition, we will provide a table titled "Research map" to visualize the status of current research and frequently used outcomes to assist with future research planning. The findings of our study may help researchers and practitioners identify knowledge gaps in the literature regarding acupuncture for PTSD.

### Ethics and dissemination

This study does not require ethical approval because we will retrieve and analyze data from previously published studies, in which informed consent was obtained by the primary investigators. This scoping review will be published in a peer-reviewed journal.

### Discussion

PTSD due to active military combat trauma is very common among military personnel and veterans [2–6]. However, PTSD treatments have many limitations [15–22]. As an alternative, complementary and integrative medicine, including acupuncture, may be considered for the treatment of PTSD [23]. Acupuncture, such as BFA, is widely used for soldiers [27,28]. To date, there have been no comprehensive reviews of acupuncture therapy for PTSD in military personnel and veterans. Therefore, we will perform the first scoping review. Our review will summarize the treatment regimens, participant characteristics, and frequently used outcomes, and will also identify knowledge gaps to assist in the planning of future studies. Because acupuncture is a non-psychological and non-pharmacological intervention, it may help overcome barriers to PTSD treatment, such as stigma in the military culture, high costs, and low

accessibility to mental health care [37]. The fundamental data obtained in this scoping review will contribute to the designing of future studies that aim to collect evidence on the effectiveness and safety of acupuncture-based treatments for PTSD in military veterans. Furthermore, the findings from this review will also be helpful to clinicians who want to use acupuncture to manage PTSD in this population. Nevertheless, this study has some limitations. First, consultation, which is both the last step of scoping registration and an optional sixth step, cannot be planned. This is because it is difficult and unethical to verify the results of this review by creating an artificial war environment for patients with PTSD. Second, although we will perform a comprehensive search, related studies published in languages other than English, Korean, Chinese, and Japanese may be excluded.

For further dissemination, the research findings will be submitted to an appropriate scientific journal. This study will also be presented at a neuropsychiatric academic conference. Moreover, we will develop an e-leaflet to provide the key findings of our review to the research community via social network services.

## Conclusions

The findings of this scoping review will provide fundamental data that will help researchers identify appropriate research questions and design further studies on the use of acupuncture for PTSD management in military veterans. These results will be helpful for developing disaster site-specific research protocols for future clinical trials on this topic.

## Supporting information

**S1 Checklist. PRISMA-P (Preferred Reporting Items for Systematic review and Meta-Analysis Protocols) 2015 checklist: Recommended items to address in a systematic review protocol\*.**
(DOC)

**S1 Table. Search terms used in each database.**
(DOCX)

## Author Contributions

**Conceptualization:** Jungtae Leem, Sang-Ho Kim.

**Data curation:** Hye-Bin Seung, Hui-Yong Kwak.

**Investigation:** Hye-Bin Seung, Hui-Yong Kwak.

**Methodology:** Chan-Young Kwon.

**Software:** Hui-Yong Kwak.

**Supervision:** Sang-Ho Kim.

**Writing – original draft:** Hye-Bin Seung, Jungtae Leem.

**Writing – review & editing:** Hui-Yong Kwak, Chan-Young Kwon, Sang-Ho Kim.

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
