## [Decision Letter · Decision Letter 0]

2 Oct 2022

PONE-D-22-21401

Acupuncture for military veterans with posttraumatic stress disorder and related symptoms after combat exposure: Protocol for a scoping review of clinical studies

PLOS ONE

Dear Dr. Kim,

Thank you for submitting your manuscript to PLOS ONE. After careful consideration, we feel that it has merit but does not fully meet PLOS ONE’s publication criteria as it currently stands. Therefore, we invite you to submit a revised version of the manuscript that addresses the points raised during the review process.

We look forward to receiving your revised manuscript.

Kind regards,

Juan-Luis Castillo-Navarrete, Ph.D.

Academic Editor

PLOS ONE

Journal Requirements:

   "This work was supported by the National Research Foundation of Korea (NRF) grant funded by the Korean government (MSIT) (No. 2021R1F1A105928211). SHK have received this fund. The funding source had no input in the interpretation or publication of the study results. ※ MSIT: Ministry of Science and ICT" 

   "The authors of this work have nothing to disclose." 

4. We note that this manuscript is a systematic review or meta-analysis; our author guidelines therefore require that you use PRISMA guidance to help improve reporting quality of this type of study. Please upload copies of the completed PRISMA checklist as Supporting Information with a file name “PRISMA checklist”.

Additional Editor Comments:

In relation to the work, which is very thorough, there are minor revisions and adjustments that need to be made.  These are indicated by the reviewers' comments. Please do not consider the indications raised by a reviewer in relation to the delivery of results and analysis of these. Now, if there are preliminary results or an approximation to them, it will be very convenient to include them.

Reviewers' comments:

Reviewer's Responses to Questions

**Comments to the Author**

1. Does the manuscript provide a valid rationale for the proposed study, with clearly identified and justified research questions?

Reviewer #1: Yes

Reviewer #2: Partly

2. Is the protocol technically sound and planned in a manner that will lead to a meaningful outcome and allow testing the stated hypotheses?

Reviewer #1: Yes

Reviewer #2: Partly

3. Is the methodology feasible and described in sufficient detail to allow the work to be replicable?

Reviewer #1: Yes

Reviewer #2: No

4. Have the authors described where all data underlying the findings will be made available when the study is complete?

Reviewer #1: Yes

Reviewer #2: Yes

5. Is the manuscript presented in an intelligible fashion and written in standard English?

Reviewer #1: Yes

Reviewer #2: Yes

6. Review Comments to the Author

You may also provide optional suggestions and comments to authors that they might find helpful in planning their study.

Reviewer #1: - Should include Web of Science and Scopus databases.

- “We will also consider gray literature searches using Google Scholar” → Be specific on the types of documents that will be used in gray literature.

- “as well studies that did not use diagnostic criteria” → At least, medical diagnostic will be the minimum to include cases.

- “With the exception of East Asian traditional medicine interventions, such as herbal medicine, moxibustion, cupping, and tui-na, any type of control group intervention will be included.” → Explain this exclusion.

- Update the PRISMA flowchart to 2020 version.

Reviewer #2: Recommendations for authors:

Revise verb tenses in the abstract.

There are several parts in the manuscript that require english review.

The abstract is expected to describe the main results of the study.

The conclusion presented in the abstract is not specific to the topic of this study, i.e., Acupuncture for military veterans with posttraumatic stress 2 disorder and related symptoms after combat exposure. The current conclusion is very generic, and could even be used as a template for any other study, it does not provide a solid conclusion derived from the subject of the review.

Regarding the research questions the authors could have incorporated: some questions about (a) theoretical aspects of the research; (b) limitations reported by previous studies, (c) aspects referred to the measurement instruments; (d) the effect size of the interventions. All these questions would be important to incorporate, in order to advance from only a descriptive component of the review and to be able to carry out analyses, for example, of the effectiveness of the interventions. The above, considering the objective specified by the authors in L114 is "We aim to explore the clinical research designs that have been used in studies conducted on acupuncture treatment for war veterans with PTSD." In addition, these questions should be incorporated because the authors state in L164 "Eligibility criteria: study types", that they will especially consider: "Clinical research studies examining the effects of acupuncture in patients with PTSD (military 166 veterans) will be included (see in L165 and L166).

The method does not consider the reference databases Wos and Scopus.

In the method, specifically in the phase "Stage 2: Identifying relevant studies" (L147) "Information sources" (L148), the authors indicate in lines 156 and L157, that they will consider "The reference lists of the relevant systematic reviews and retrieved articles will be manually searched". At this point, a concern arises about the originality of the work. They should include in the introduction the previous existing systematic reviews on the subject and perhaps make a table with the following information: citation, objective of the previous systematic review, keywords, search period, databases consulted and main results. This would allow understanding the value of the review by clearly identifying the contributions that the authors will make in relation to the objectives and findings of previous reviews. However, in the discussion of the manuscript the authors point out that there are no reviews of this type. This generates an inconsistency in the arguments (see L248 and L249) "there have been no comprehensive reviews of acupuncture therapy for PTSD in military personnel and veterans. Therefore, we will perform the first scoping review."

In line 162 the authors point out that there is a supplementary material with "The search terms and strategies are detailed in S1 Supplementary digital content". When downloading this file, in the search algorithm tables, there are only the keywords with their synonyms and the respective booleans used, e.g. OR, AND etc., but there is no syntax for each search, i.e. the filters applied in each database, e.g. disciplinary area. Finally, it is also not specified whether iterations will be performed in each database and with which different combination. This biases the possibility of replicating the study; the keywords and year of search are not enough, since there are multiple possible combinations that are not clarified by the authors.

Figure 1 is empty. But, in addition, it is not the latest version of the PRISMA RSL process which were the guidelines on which the authors based themselves as stated in the method. The Flowchart acts only has 3 phases, not 5. Therefore the method is outdated (authors are suggested to revise. https://www.prisma-statement.org//PRISMAStatement/FlowDiagram

It is necessary to present the results of the review with an exhaustive discussion of each one of them. Only describing a protocol with important limitations that do not allow its replicability is insufficient to contribute to the generation of knowledge in the area.

The authors need to make a profound change in the presentation of their manuscript for it to be considered for publication. In its current state, it does not have sufficient complexity and scientific novelty.

7. PLOS authors have the option to publish the peer review history of their article (what does this mean?). If published, this will include your full peer review and any attached files.

Reviewer #1: No

Reviewer #2: **Yes: **Fabiola Sáez Delgado

---

## [Author Response · Author response to Decision Letter 0]

24 Oct 2022

Response to the Reviewers’ Comments

Title: Acupuncture for military veterans with posttraumatic stress disorder and related symptoms after combat exposure: Protocol for a scoping review of clinical studies

Emily Chenette 

Editor-in-Chief

PLOS ONE

Dear Editor Juan-Luis Castillo-Navarrete, Ph.D:

I wish to re-submit the manuscript titled “Acupuncture for military veterans with posttraumatic stress disorder and related symptoms after combat exposure: Protocol for a scoping review of clinical studies.” The manuscript ID is PONE-D-22-21401.

We thank you and the reviewers for your thoughtful suggestions and insights. The manuscript has benefited from these insightful suggestions. I look forward to working with you and the reviewers to move this manuscript closer to publication in the PLOS ONE.

The manuscript has been rechecked and the necessary changes have been made in accordance with the reviewers’ suggestions. The responses to all comments have been prepared and attached herewith/given below. 

Thank you for your consideration. I look forward to hearing from you.

Response to Comments from the academic editor 

Comment #1: 

Response #1:

Thank you for your comment. I have reviewed manuscript carefully to meet style requirements. I changed some parts (marked in yellow).

I delete keywords. And I moved Ethics and dissemination to methods section. 

“The search terms and strategies are detailed in S1 Table.”

“Conclusions”

“Supporting information

S1 Table. Search terms used in each database.

S2 Table. PRISMA-P (Preferred Reporting Items for Systematic review and Meta-Analysis Protocols) 2015 checklist: recommended items to address in a systematic review protocol*

Comment #2: 

Thank you for stating the following financial disclosure: 

 "This work was supported by the National Research Foundation of Korea (NRF) grant funded by the Korean government (MSIT) (No. 2021R1F1A105928211). SHK has received this fund. The funding source had no input in the interpretation or publication of the study results. ※ MSIT: Ministry of Science and ICT" 

Response #2:

I added the funders’ role in the financial disclosure.

“This work was supported by the National Research Foundation of Korea (NRF) grant funded by the Korean government (MSIT) (No. 2021R1F1A105928211). SHK has received this fund. The funding source had no input in the interpretation or publication of the study results. The funders had no role in study design, data collection and analysis, decision to publish, or preparation of the manuscript.”

Comment #3: 

Thank you for stating the following in your Competing Interests section: 

 "The authors of this work have nothing to disclose." 

Response #3:

I changed the statement of Competing Interests. I changed the previous statement to this in the cover letter.

[cover letter] “This manuscript has not been published or presented elsewhere in part or in entirety and is not under consideration by another journal. No ethical approval is needed for our manuscript because data from previously published studies, in which informed consent was obtained by the primary investigators, will be retrieved and analyzed. We have read and understood your journal’s policies, and we believe that neither the manuscript nor the study violates any of these. The authors have declared that no competing interests exist.”

Comment #4: 

We note that this manuscript is a systematic review or meta-analysis; our author guidelines therefore require that you use PRISMA guidance to help improve reporting quality of this type of study. Please upload copies of the completed PRISMA checklist as Supporting Information with a file name “PRISMA checklist”.

Response #4:

I provided PRISMA-P (Preferred Reporting Items for Systematic review and Meta-Analysis Protocols) 2015 checklist: recommended items to address in a systematic review protocol*as Supporting Information (S2 table).

Comment #5: 

Response #5:

I have checked all reference lists. There are no retracted papers. I also have corrected information of some references such as issue and page. 

Comment #6: Additional Editor Comments:

In relation to the work, which is very thorough, there are minor revisions and adjustments that need to be made. These are indicated by the reviewers' comments. Please do not consider the indications raised by a reviewer in relation to the delivery of results and analysis of these. Now, if there are preliminary results or an approximation to them, it will be very convenient to include them.

Response #6:

Thank you for your careful comment. I prepared the responses to all comments from reviewers as below.

Response to Comments from Reviewer 1

Comment #1: 

Should include Web of Science and Scopus databases.

Response #1:

According to your suggestion, I added Wos and Scopus to the searching database of abstract, method section, and Search terms used in each database (S1 Table).

Comment #2: 

“We will also consider gray literature searches using Google Scholar” → Be specific on the types of documents that will be used in gray literature.

Response #2:

We will also consider gray literature searches such as conference proceedings and doctoral dissertations using Google Scholar.[Page 8]

Comment #3: 

“as well studies that did not use diagnostic criteria” → At least, medical diagnostic will be the minimum to include cases.

Response #3:

I changed those sentences for clarity as below:

“We will include studies that used standardized diagnostic criteria for PTSD (such as those presented in the Diagnostic and Statistical Manual of Mental Disorders and the International Classification of Diseases). We will also include studies that used cut-off values from validated PTSD evaluation tools (such as the PTSD Checklist, Impact of Event Scale-Revised, and Clinician-Administered PTSD Scale) as the inclusion criteria for participants.” [Page 9]

Comment #4: 

“With the exception of East Asian traditional medicine interventions, such as herbal medicine, moxibustion, cupping, and tui-na, any type of control group intervention will be included.” → Explain this exclusion.

Response #4:

We want to compare the effectiveness of acupuncture for PTSD management in military veterans with usual conventional therapy (medication or psychotherapy) or sham-acupuncture or wait-list group. Therefore, we have excluded East Asian traditional medicine interventions such as herbal medicine, moxibustion, cupping, and tui-na. 

Comment #5: 

 Update the PRISMA flowchart to 2020 version.

Response #5:

I have updated the PRISMA flowchart to 2020 version (S1 Fig) as your comment.

Response to Comments from Reviewer 2

Comment #1: 

#1. Revise verb tenses in the abstract.

Response #1:

I revised verb tenses as bellows (marked in yellow):

“Methods: ….The study question is as follows…”

Comment #2: 

2. There are several parts in the manuscript that require english review.

Response #2:

I have attached the certification of editing for manuscript. I have received extensive editing again for this revision. 

Comment #3: 

#3. The abstract is expected to describe the main results of the study. The conclusion presented in the abstract is not specific to the topic of this study, i.e., Acupuncture for military veterans with posttraumatic stress 2 disorder and related symptoms after combat exposure. The current conclusion is very generic, and could even be used as a template for any other study, it does not provide a solid conclusion derived from the subject of the review.

Response #3: 

Thank you for insightful comment. According to your suggestion, I did re-write the conclusion.

“The findings of this scoping review will provide fundamental data that will help researchers identify appropriate research questions and design further studies on the use of acupuncture for PTSD management in military veterans. These results will be helpful for developing disaster site-specific research protocols for future clinical trials on this topic. (marked in yellow).” [page 3, 13] 

Comment #4: 

#4. Regarding the research questions the authors could have incorporated: some questions about (a) theoretical aspects of the research; (b) limitations reported by previous studies, (c) aspects referred to the measurement instruments; (d) the effect size of the interventions. All these questions would be important to incorporate, in order to advance from only a descriptive component of the review and to be able to carry out analyses, for example, of the effectiveness of the interventions. The above, considering the objective specified by the authors in L114 is "We aim to explore the clinical research designs that have been used in studies conducted on acupuncture treatment for war veterans with PTSD." In addition, these questions should be incorporated because the authors state in L164 "Eligibility criteria: study types", that they will especially consider: "Clinical research studies examining the effects of acupuncture in patients with PTSD (military 166 veterans) will be included (see in L165 and L166).

Response #4:

I have incorporated previous research questions as below (marked in yellow):

1. Which clinical research designs were adopted in studies on acupuncture for PTSD management in military veterans?

2. What is the most frequently used type of acupuncture therapy for PTSD management in military veterans? 

3. Which clinical outcomes were adopted in previous studies on PTSD management in military veterans? 

4. What types of adverse events occurred after acupuncture therapy for PTSD in military veterans?

5. How long should acupuncture treatment be administered for PTSD management? 

6. Which populations were the target in previous PTSD acupuncture studies?

=>

1. What are the characteristics (e.g., research design and target population) of studies on the use of acupuncture for PTSD management in military veterans?

2. Which clinical outcomes were adopted in previous studies on PTSD management in military veterans? 

3. What is the regimen of acupuncture therapy for PTSD management in military veterans? 

4. What have previous studies reported on with respect to the effectiveness and safety of using acupuncture for treating PTSD in military veterans? [page 7]

Comment #5: 

#5. The method does not consider the reference databases Wos and Scopus.

Response #5:

According to your suggestion, I added Wos and Scopus in the searching database of abstract and method section (marked in yellow).

Information sources

We will restrict this review to peer-reviewed studies on acupuncture for PTSD. A literature search will be conducted from inception to June 2022. The following databases will be searched: Medline (via PubMed), Excerpta Medica dataBASE, Cochrane Central Register of Controlled Trials, Web of Science, Scopus, Allied and Complementary Medicine Database, [page 8]

Comment #6: 

#6. In the method, specifically in the phase "Stage 2: Identifying relevant studies" (L147) "Information sources" (L148), the authors indicate in lines 156 and L157, that they will consider "The reference lists of the relevant systematic reviews and retrieved articles will be manually searched". At this point, a concern arises about the originality of the work. They should include in the introduction the previous existing systematic reviews on the subject and perhaps make a table with the following information: citation, objective of the previous systematic review, keywords, search period, databases consulted and main results. This would allow understanding the value of the review by clearly identifying the contributions that the authors will make in relation to the objectives and findings of previous reviews. However, in the discussion of the manuscript the authors point out that there are no reviews of this type. This generates an inconsistency in the arguments (see L248 and L249) "there have been no comprehensive reviews of acupuncture therapy for PTSD in military personnel and veterans. Therefore, we will perform the first scoping review."

Response #6:

To our knowledge, there are no comprehensive reviews regarding this topic. I changed those sentences for clarity as below (marked in yellow):

The reference lists of the relevant reviews and retrieved articles will be manually searched". [page 8]

Comment #7: 

#7. In line 162 the authors point out that there is a supplementary material with "The search terms and strategies are detailed in S1 Supplementary digital content". When downloading this file, in the search algorithm tables, there are only the keywords with their synonyms and the respective booleans used, e.g. OR, AND etc., but there is no syntax for each search, i.e. the filters applied in each database, e.g. disciplinary area. Finally, it is also not specified whether iterations will be performed in each database and with which different combination. This biases the possibility of replicating the study; the keywords and year of search are not enough, since there are multiple possible combinations that are not clarified by the authors.

Response #7:

Thank you for your comment. I added syntax to each search terms of EMBASE and PsycARTICLES (marked in yellow). 

Comment #8: 

#8. Figure 1 is empty. But, in addition, it is not the latest version of the PRISMA RSL process which were the guidelines on which the authors based themselves as stated in the method. The Flowchart acts only has 3 phases, not 5. Therefore the method is outdated (authors are suggested to revise. https://www.prisma-statement.org//PRISMAStatement/FlowDiagram

Response #8:

I have updated the PRISMA flowchart to 2020 version (S1 Fig) as your comment.

Comment #9: 

#9. It is necessary to present the results of the review with an exhaustive discussion of each one of them. Only describing a protocol with important limitations that do not allow its replicability is insufficient to contribute to the generation of knowledge in the area.

Response #9:

Discussion

PTSD due to active military combat trauma is very common among military personnel and veterans [2–6]. However, PTSD treatments have many limitations [15–22]. As an alternative, complementary and integrative medicine, including acupuncture, may be considered for the treatment of PTSD [23]. Acupuncture, such as BFA, is widely used for soldiers [27,28]. To date, there have been no comprehensive reviews of acupuncture therapy for PTSD in military personnel and veterans. Therefore, we will perform the first scoping review. Our review will summarize the treatment regimens, participant characteristics, and frequently used outcomes, and will also identify knowledge gaps to assist in the planning of future studies Because acupuncture is a non-psychological and non-pharmacological intervention, it may help overcome barriers to PTSD treatment, such as stigma in the military culture, high costs, and low accessibility to mental health care [37]. The fundamental data obtained in this scoping review will contribute to the designing of future studies that aim to collect evidence on the effectiveness and safety of acupuncture-based treatments for PTSD in military veterans. Furthermore, the findings from this review will also be helpful to clinicians who want to use acupuncture to manage PTSD in this population.[Page 12,13]

37. Bisson JI, van Gelderen M, Roberts NP, Lewis C. Non-pharmacological and non-psychological approaches to the treatment of PTSD: results of a systematic review and meta-analyses. European Journal of Psychotraumatology. 2020 Dec 31;11(1):1795361.

Comment #10: 

#10. The authors need to make a profound change in the presentation of their manuscript for it to be considered for publication. In its current state, it does not have sufficient complexity and scientific novelty.

Response #10:

This study followed the existing scoping review protocol methodology. However, since the subject of this review is a new topic that has not been studied, we hope that the results of review will contribute to conducting novel study in the future.

---

## [Editor Report · Decision Letter 1]

27 Oct 2022

Acupuncture for military veterans with posttraumatic stress disorder and related symptoms after combat exposure: Protocol for a scoping review of clinical studies

PONE-D-22-21401R1

Dear Dr. Sang-Ho Kim,

We’re pleased to inform you that your manuscript has been judged scientifically suitable for publication and will be formally accepted for publication once it meets all outstanding technical requirements.

Kind regards,

Juan-Luis Castillo-Navarrete, Ph.D.

Academic Editor

PLOS ONE

Additional Editor Comments (optional):

Considering the interesting and novelty of the study protocol, we look forward to the results obtained, which we hope will also be published in PLOS ONE.
---

## [Editor Report · Acceptance letter]

14 Apr 2023

PONE-D-22-21401R1 

Acupuncture for military veterans with posttraumatic stress disorder and related symptoms after combat exposure: Protocol for a scoping review of clinical studies 

Dear Dr. Kim:

I'm pleased to inform you that your manuscript has been deemed suitable for publication in PLOS ONE. Congratulations! Your manuscript is now with our production department. 

Kind regards, 

on behalf of

Dr. Juan-Luis Castillo-Navarrete 

Academic Editor

PLOS ONE